# A Tree-Structured Multi-Task Model Recommender

**Lijun Zhang**[1]  **Xiao Liu**[1]  **Hui Guan**[1]

[1]College of Information and Computer Sciences, University of Massachusetts Amherst

**Abstract**  Tree-structured multi-task architectures have been employed to jointly tackle multiple vision tasks in the context of multi-task learning (MTL). The major challenge is to determine where to branch out for each task given a backbone model to optimize for both task accuracy and computation efficiency. To address the challenge, this paper proposes a recommender that, given a set of tasks and a convolutional neural network-based backbone model, automatically suggests tree-structured multi-task architectures that could achieve a high task performance while meeting a user-specified computation budget without performing model training. Extensive evaluations on popular MTL benchmarks show that the recommended architectures could achieve competitive task accuracy and computation efficiency compared with state-of-the-art MTL methods. Our tree-structured multi-task model recommender is open-sourced and available at `https://github.com/zhanglijun95/TreeMTL`.

## 1 Introduction

Multi-task learning (MTL) aims to solve multiple tasks simultaneously. Compared to independently learning tasks, it is an effective approach to improve task performance while reducing computation and storage costs. However, over-sharing information between tasks can cause task interference (Sener and Koltun, 2018; Maninis et al., 2019) and accuracy degradation. The major challenge in designing a multi-task architecture is thus to identify an intermediate state between over-shared and independent architectures (i.e., a partially-shared architecture), which not only preserves the benefits of lower computation cost and memory overhead, but also avoid task interference as much as possible to guarantee acceptable task accuracy. Such a partially-shared architecture is also called *a tree-structured multi-task architecture*. Its shallow network layers are shared across tasks like tree roots, whereas deeper ones gradually grow more task-specific like tree branches (Vandenhende et al., 2019). Identifying the best tree-structured multi-task architecture needs to determine where to branch out for each task to optimize for both computation efficiency and task accuracy.

Previous works opted for the simplest strategy of sharing the initial layers of a backbone model, after which all tasks branch out simultaneously (Ruder, 2017; Nekrasov et al., 2019; Suteu and Guo, 2019; Leang et al., 2020). Since the point at which the branching occurs is determined manually, they call for domain expertise when tackling different tasks and usually result in unsatisfactory solutions due to the enormous architecture design space. To automate architecture design, one line of work deduced the layer sharing possibility based on measurable task relatedness (Lu et al., 2017; Vandenhende et al., 2019; Standley et al., 2020) and minimized the total task dissimilarity when designing multi-task architectures. However, they ignore task interactions that could bring the potential generalization improvement and positive inhibition of overfitting when multiple tasks are trained together (Ruder, 2017; Vandenhende et al., 2020). Another line of work attempted to learn how to branch a network such that the overall multi-task loss is minimized via differentiable neural architecture search (Bruggemann et al., 2020; Guo et al., 2020). Such end-to-end frameworks integrated the architecture search with the network training process, which easily leads to sub-optimal multi-task architectures (Choromanska et al., 2015; Sun et al., 2020) due to training difficulties. Besides, the learned multi-task architectures cannot guarantee to meet a user-defined computation budget since these methods are like a black box where users cannot control the exploring process.

In this paper, we overcome the aforementioned limitations and propose a tree-structured multi-task model recommender. It takes as inputs an arbitrary convolutional neural network (CNN) backbone model and a set of tasks in interest, and then predicts the top-$k$ tree-structured multi-task architectures that achieve high task accuracy while meeting a user-specified computation budget. Our basic idea is to build a *task accuracy estimator* that can predict the task accuracy of each multi-task model architecture in the design space without performing model training. The task accuracy estimator captures task interactions by leveraging the task performance of well-trained two-task architectures instead and enables ranking of all multi-task architectures with more than two tasks using their predicted task accuracy. The recommender can then *enumerate the design space* and identify the multi-task models with the highest predicted task accuracy. Unlike differentiable neural architecture search-based approaches, the recommender is a white-box that allows users to easily control the computation complexity of the multi-task architectures. The basic idea poses three major research questions:

- **RQ 1**: how to build an *accurate* task accuracy estimator that enables a faithful ranking of the multi-task architectures in the design space based on their estimated task performance?
- **RQ 2**: how to *represent* multi-task model architectures such that a recommender can *completely* enumerate the design space for estimating their performance?
- **RQ 3**: how to *automatically* support various CNN backbone models?

To answer **RQ 1**, our task accuracy estimator predicts the task accuracy of a multi-task architecture by averaging the task accuracy of associated well-trained two-task architectures. A ranking score of the multi-task architecture is calculated as the weighted sum of the tasks' accuracy, where the weight of each task is determined by quantified accuracy variance to ensure faithful ranking. To answer **RQ 2**, we propose a novel data structure called *Layout* to represent a multi-task architecture and an operation called *Layout Cut* to derive multi-task architectures. We further propose a *cut-based recursive algorithm* that is proved to be able to enumerate the design space completely. To answer **RQ 3**, we design a *branching point detector* to automatically separate a CNN backbone model into a sequence of computation blocks where each block corresponds to a possible branching point.[1] The detector saves manual efforts in applying the recommender to an arbitrary CNN architecture.

Experiments on popular MTL benchmarks, NYUv2 (Silberman et al., 2012) and Tiny-Taskonomy (Zamir et al., 2018), using different backbone models, Deeplab-ResNet34 (Chen et al., 2017) and MobileNetV2 (Sandler et al., 2018), demonstrate that the recommended tree-structured multi-task architectures achieve competitive task accuracy compared with state-of-the-art MTL methods under specified computation budgets. Our empirical evaluation also demonstrates that ranking of the multi-task architectures using estimated task accuracy without training has a high correlation (Pearson's $\gamma$ is $0.5 \sim 0.85$) with the oracle ranking after training for different CNN architectures.

## 2 Related Works

Multi-task learning (MTL) is commonly categorized into either hard or soft parameter sharing (Ruder, 2017; Vandenhende et al., 2020). In hard parameter sharing, a set of parameters in the backbone model are shared among tasks. In soft parameter sharing (Misra et al., 2016; Ruder et al., 2019; Gao et al., 2019), each task has its own set of parameters. Task information is shared by applying regularization on parameters during training, such as enforcing the weights of the model for each task to be similar. In this paper, we focus on hard parameter sharing as it produces memory- and computation-efficient multi-task models.

---

[1]A branching point usually corresponds to a micro-architecture such as a residual block in ResNet50, following prior works (Vandenhende et al., 2019; Guo et al., 2020; Bruggemann et al., 2020).

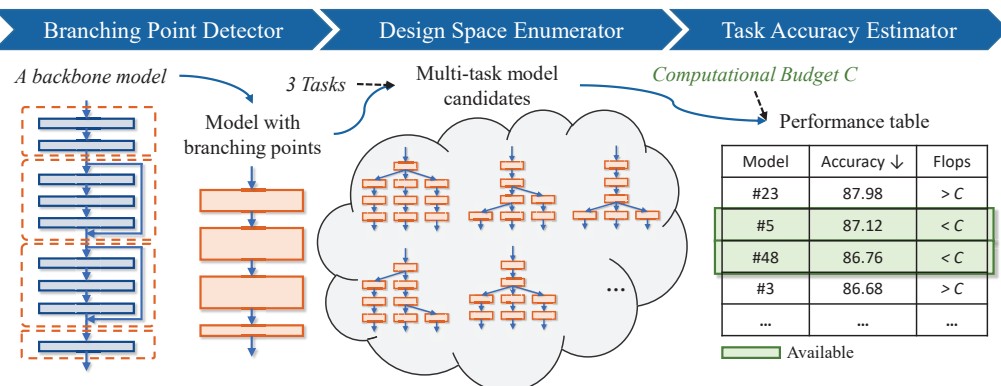

Figure 1: Tree-structured multi-task model recommender workflow.

Early works on multi-task architecture design rely on domain expertise to decide which layers should be shared across tasks and which ones should be task-specific (Long et al., 2017; Nekrasov et al., 2019; Suteu and Guo, 2019; Leang et al., 2020). Due to the enormous design space, such approaches are difficult to find an optimal solution.

In recent years, researchers attempt to automate the procedure of designing multi-task architectures. Deep Elastic Network (DEN) (Ahn et al., 2019) uses reinforcement learning (RL) to determine whether each filter in convolutional layers can be shared across tasks. Similarly, AdaShare (Sun et al., 2019) and AutoMTL (Zhang et al., 2021) learn task-specific policies that select which layers to execute for a given task. Some other works (Gao et al., 2020; Wu et al., 2021) adopt NAS techniques to explore feature fusion opportunities across tasks. Their primary goal is to improve task accuracy instead of computation efficiency by minimizing the overall multi-task loss. Thus there is no guarantee that the searched multi-task model architectures will meet the computation budget. Also, their architecture search procedure requires substantial search time and is usually hard to converge since the sharing strategy and network parameters generally prefer the alternating training principle to stabilize the training process (Xie et al., 2018; Sun et al., 2019; Wu et al., 2019).

Our work pays more attention to balancing task accuracy and computation efficiency through recommending branching structures for multi-task models. There also exist several interesting methods under this direction. FAFS (Lu et al., 2017) starts from a thin network where tasks initially share all layers and dynamically grows the model in a greedy layer-by-layer fashion depending on task similarities. It computes task similarity based on the likelihood of input samples having the same difficulty level. What-to-Share (Vandenhende et al., 2019) measures the task affinity by analyzing the representation similarity between independent models for each task. It recommends the multi-task architecture with the minimum total task dissimilarity. However, because the task dissimilarity between two tasks is always non-negative, the theoretical optimal multi-task architecture would be always independent models whose total task dissimilarity is zero. In contrast to pre-computing the task relatedness, BMTAS (Bruggemann et al., 2020) and Learn-to-Branch (Guo et al., 2020) utilize differentiable neural architecture search to construct end-to-end trainable frameworks that integrate the architecture exploration with the network training process. These learning-based methods easily lead to the suboptimal multi-task model (Choromanska et al., 2015; Sun et al., 2020) due to difficulties in training and cannot guarantee the resulting multi-task architecture to obey a user-defined computation budget.

## 3 Proposed Approach

Given a backbone model with $B$ branching point and a set of $T$ tasks, our goal is to build a recommender that, when deployed, predicts $k$ tree-structured multi-task architectures that achieve a high task accuracy while meeting a user-specified computation budget $C$. Figure 1 illustrates

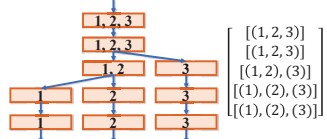 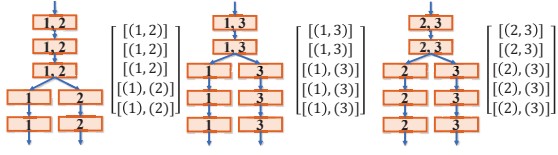

(a) A multi-task model for three tasks.  (b) Associated two-task models.

Figure 2: A multi-task architecture and the related two-task architectures. The average of task accuracy in (b) is a good indicator of the task accuracy in (a).

the offline building process and the online usage of the recommender. During the offline building process, users provide an arbitrary CNN-based backbone model and a set of tasks. A *branching point detector* will automatically identify the sequential computational blocks in the backbone model, each of the blocks corresponding to a viable branching point. A *task accuracy estimator* is then built based on the given set of tasks and the identified branching points to predict the performance of all the tree-structured multi-task architectures in the design space, which are explored using a *design space enumerator*. The performance of these multi-task architectures including their other attributes such as model size, FLOPs etc. could be stored in a performance table to facilitate online queries. When deployed, the recommender takes a user-specified computation budget $C$ as input, and suggests multi-task architectures by looking up the performance table on the fly. We next elaborate the three major components, *task accuracy estimator*, *design space enumerator*, and *branching point detector* in detail.

### 3.1 Task Accuracy Estimator

Task accuracy estimator predicts the task accuracy of a tree-structured multi-task architecture without performing actual model training. The problem is challenging because predicting a single-task architecture's accuracy is already non-trivial and multi-task architectures introduce more complexities due to task interactions and interference. Task accuracy estimator addresses the problem by leveraging well-trained two-task architectures to quantify task accuracy and interactions and predict the performance of a multi-task architecture. Specifically, for any tree-structured multi-task architecture, the estimator predicts its task accuracy by averaging the task accuracy of all the *associated* two-task architectures. A two-task architecture is considered associated if it meets both conditions: (1) the two tasks are a subset of the tasks in the multi-task architecture; (2) the two tasks have the identical branching point as in the multi-task architecture.

Figure 2 illustrate the basic idea. Figure 2(a) shows a multi-task architecture constructed from a backbone model with five branching points for three tasks and Figure 2(b) shows the three associated two-task architectures. The numbers inside each block indicate among which tasks the block is shared. Tasks 1 and 2 branch out after the third block, which is the same branching point as the first two-task architecture in Figure 2(b). Similarly, tasks 1 and 3 branch out after the second block, which is the same branching point as the second two-task architecture in Figure 2(b). We estimate task 1 accuracy of the multi-task architecture by averaging task 1 accuracy of the first and second two-task architectures, task 2 accuracy from those of the first and third two-task models, and task 3 accuracy from those of the second and third two-task models.

The ultimate goal of the task accuracy estimator is to enable ranking of multi-task architectures based on their estimated task accuracy. Due to the noise in training two-task architectures (Pham et al., 2020), an estimated task accuracy of a multi-task architecture could suffer from some accuracy variance and lead to an inaccurate ranking. A *ranking score* is thus calculated as the *weighted* sum of the tasks' performance. Tasks with higher accuracy variance have lower *task weight*.

To quantify accuracy variance and task weight, we adopt the Singular Value Decomposition Entropy (SVDE) (Li et al., 2008; Jelinek et al., 2019) to measure the regularity of each task $t_i$'s performance in its $B + 1$ two-task architectures with another task $t_j$. SVDE has two important features that make it suitable for regularity measurement. (1) SVDE is scale-invariant — that is,

it characterizes the accuracy trend rather than the absolute values of the accuracy sequence. (2) SVDE is order-aware since it is designed for data sequences, not just for a set of data. The order of data items affects data regularity, which is captured by SVDE. SVDE reflects the number of orthogonal vectors contributed to a task performance sequence $(\Delta t_i^{(0)}, \ldots, \Delta t_i^{(B)} | t_i, t_j)$, where $\Delta t_i^{(b)}$ is $t_i$'s performance in a two-task model that branches at $b$-th point. Higher entropy indicates lower regularity and thus higher variance. The *task weight* of $t_i$ is the average of the negative entropy over all possible two task combinations $(t_i, t_j), \forall j \neq i$:

$$w_i = \frac{1}{T-1} \sum_{j \in \mathcal{T}, j \neq i} -SVDE(\Delta t_i^{(0)}, \ldots, \Delta t_i^{(B)} | t_i, t_j), \tag{1}$$

where $\mathcal{T}$ is the set of tasks and $T = |\mathcal{T}|$ is the number of tasks. The final ranking score of a multi-task architecture is:

$$S = normalize([w_1, \ldots, w_T])^T [\Delta t_1, \ldots, \Delta t_T], \tag{2}$$

where $normalize([w_1, \ldots, w_T])$ is the normalized task weights so that their sum is equal to one.

Estimating the accuracy of a multi-task architecture requires training of its associated two-task architectures. Given $T$ number of tasks and $B$ number of branching points, the total number of two-task architectures to train is $C_T^2 \cdot (B+1)$, where $C_T^2$ is the number of two task combinations. The training overhead of the two-task architectures is much less than training all the multi-task architectures whose number is $O([(2^{T-1} - 1) \cdot B]^{T-1})$. Our experiments in Section 4.3 demonstrate that the ranking of multi-task architectures using estimated task accuracy without training has a high correlation (Pearson's $\gamma$ is $0.5 \sim 0.85$) with the oracle ranking from training for different CNN architectures.

### 3.2 Design Space Enumerator

Design space enumerator formalizes the representation of tree-structured multi-task architectures so that the recommender can completely enumerate the design space. It introduces a data structure called *Layout* and an operator called *Layout Cut* to derive multi-task architectures. Based on the two core concepts, we propose a cut-based recursive algorithm to enumerate all possible architectures.

**Definition 3.1** (Layout). *A layout is a symbolized representation of a tree-structured multi-task architecture. Formally, for $T$ tasks and a backbone model with $B$ branching points, a layout $\mathbf{L} = [L_1, L_2, \cdots, L_B]$, where $L_i$ is a list of task sets at the $i$-th branching point. Task sets in $L_i = [L_i^1, L_i^2, \cdots]$ are subsets of tasks $\mathcal{T}$ and satisfy two conditions: (1) $L_i^1 \cup L_i^2 \cup \cdots = \mathcal{T}$, and (2) $L_i^p \cap L_i^q = \emptyset, \forall L_i^{\{p,q\}} \in L_i$.*

A task set $L_i^p$ means the set of tasks in $L_i^p$ sharing the $i$-th block. Figure 2 illustrates the layouts of a multi-task architecture and three two-task architectures. We define the initial layout as $\mathbf{L}_0 = [\underbrace{[\mathcal{T}], \cdots, [\mathcal{T}]}_{B}] = [\underbrace{[\{t_1, \ldots, t_T\}], \cdots, [\{t_1, \ldots, t_T\}]}_{B}]$, which means all the tasks share all the blocks in the multi-task model.

**Definition 3.2** (Layout Cut). *A layout cut is an operator that transforms one layout to another layout by selecting an available task set and dividing it into two task sets.*

Based on Definition 3.2, we propose a cut-based algorithm to enumerate all possible layouts, namely tree-structured multi-task architectures. The main idea is to recursively apply layout cuts on the initial layout $\mathbf{L}_0$ and all the generated layouts until no new layout is generated.

**Theorem 3.1.** *The cut-based layout enumeration algorithm could explore the design space of tree-structured multi-task models completely.*

### 3.3 Branching Point Detector

Branching point detector allows the recommender to support an arbitrary CNN backbone model without manual reimplementation. Its design is motivated by the observation that common CNN backbone models are a sequence of *computation blocks*, such as residual blocks in ResNet50 (He et al., 2016) and bottleneck blocks in MobileNetV2 (Sandler et al., 2018). These computation blocks are typically treated as branching points in MTL (Vandenhende et al., 2019; Bruggemann et al., 2020) and satisfy two requirements. (1) They contain trainable parameters so that whether they are shared across tasks is likely to make a difference in task accuracy. (2) They are connected to each other sequentially–that is, there is no link across non-sequential blocks. The two requirements inspire us to design a two-stage branching point detector.

The first stage is to identify groups of operators called *candidate blocks* in a given backbone model. Each candidate block is a subgraph in the computation graph of the backbone model that takes only one input tensor and produces only one output tensor. The branching point detector leverages the Cut Theorem[2] in the Graph Theory to partition the original computation graph of the backbone model into candidate blocks. A subgraph can be divided into two subgraphs if the size of the minimum cut is one; otherwise, the subgraph can be no longer partitioned and is a candidate block. Because a candidate block could contain operators that have no parameters at all, the second stage is to merge candidate blocks that contain only unparameterized layers (e.g., ReLU, Pooling) and normalization layers (e.g., Batch Normalization) with adjacent candidate blocks (e.g., Convolution Layer) to generate final computation blocks (e.g., ConvBNReLU). Each computation block corresponds to a viable branching point.

The proposed branching point detector enables the recommender to automatically parse the backbone model and produce two-task architectures and multi-task architectures based on a layout. It saves manual efforts in generalizing multi-task architecture search across different backbone models. We also allow users to flexibly add or remove branching points to adjust the architecture search space.

## 4 Experiments

### 4.1 Experiment Settings

**Datasets and Tasks.** Our experiments are conducted on two popular datasets in multi-task learning, **NYUv2** (Silberman et al., 2012) and **Tiny-Taskonomy** (Zamir et al., 2018). The NYUv2 dataset consists of RGB-D indoor scene images and three tasks, 13-class semantic segmentation, depth estimation, and surface normal prediction. Tiny-Taskonomy contains indoor images and five tasks: semantic segmentation, surface normal prediction, depth estimation, keypoint detection, and edge detection. The data splits follow prior works (Sun et al., 2019; Zhang et al., 2021).

**Loss Functions and Evaluation Metrics.** In NYUv2, Semantic segmentation uses a pixel-wise cross-entropy loss for each predicted class label, and is evaluated using mean Intersection over Union and Pixel Accuracy (mIoU and Pixel Acc, the higher the better). Surface normal prediction uses the inverse of cosine similarity between the normalized prediction and ground truth, and is evaluated using mean and median angle distances between the prediction and the ground truth (the lower the better), and the percentage of pixels whose prediction is within the angles of 11.25°, 22.5° and 30° to the ground truth (Eigen and Fergus, 2015) (the higher the better). Depth estimation uses the L1 loss, and the absolute and relative errors between the prediction and the ground truth are computed (the lower the better). In Taskonomy, all the tasks are trained using the same loss as in NYUv2 and directly evaluated by the task-specific loss. Since tasks have multiple evaluation metrics and their value can also be at different scales, we compute a single **relative performance metric** following (Maninis et al., 2019; Sun et al., 2019). The overall performance is the average of

---

[2]https://en.wikipedia.org/wiki/Minimum_cut

Table 1: Performance of top-5 recommended architectures on NYUv2 using Deeplab-ResNet34.

| Model | FLOPs (%) ↓ | #Params (%) ↓ | Semantic Seg. mIoU ↑ | Pixel Acc. ↑ | $\Delta t_1$ ↑ | Surface Normal Prediction Error ↓ Mean | Median | $\theta$, within ↑ 11.25° | 22.5° | 30° | $\Delta t_2$ ↑ | Depth Estimation Error ↓ Abs. | Rel. | $\delta$, within ↑ 1.25 | $1.25^2$ | $1.25^3$ | $\Delta t_3$ ↑ | $\Delta t$ ↑ |
|---|---|---|---|---|---|---|---|---|---|---|---|---|---|---|---|---|---|---|
| Ind. Models | - | - | **26.50** | **58.20** | - | 17.70 | 16.30 | 29.40 | 72.30 | **87.30** | - | 0.62 | 0.24 | 57.80 | 85.80 | 96.00 | - | - |
| #0 | -66.67 | -66.66 | 25.23 | 57.69 | -2.8 | **17.14** | 15.15 | 35.85 | 72.20 | 85.54 | **6.0** | **0.55** | 0.23 | 63.85 | 89.38 | 97.03 | 5.8 | **3.0** |
| #45 | -36.56 | -35.45 | 25.18 | 57.36 | -3.2 | 17.26 | **14.93** | **36.33** | 72.27 | 85.16 | 6.4 | 0.58 | **0.22** | 62.70 | 88.79 | 96.93 | 5.5 | 2.9 |
| #50 | -46.87 | -46.13 | 24.72 | 56.71 | -4.6 | 17.24 | 15.13 | 32.17 | **72.66** | 85.75 | 3.6 | 0.56 | 0.23 | 63.87 | 88.72 | 96.81 | 5.6 | 1.5 |
| #37 | -34.87 | -33.70 | 26.30 | 57.94 | **-0.6** | 17.24 | 15.16 | 35.78 | 71.90 | 85.43 | 5.7 | 0.61 | **0.22** | 60.09 | 87.18 | 96.31 | 2.6 | 2.6 |
| #49 | -46.87 | -46.13 | 25.56 | 57.62 | -2.3 | 17.77 | 15.70 | 33.18 | 70.99 | 84.64 | 2.3 | **0.55** | **0.22** | **64.62** | **89.78** | **97.55** | 7.8 | 2.6 |

the relative performance over all tasks, namely $\Delta t = \frac{1}{T} \sum_{i=1} \Delta t_i$. The units for relative performance $\Delta t_i$ and $\Delta t$ are percentage (%).

**Baselines for Comparison.** Our baselines include both tree-structured MTL methods and general MTL approaches. For state-of-the-art tree-structured MTL methods, we compare with **What-to-Share**[3] (Vandenhende et al., 2019), **BMTAS**[4] (Bruggemann et al., 2020), **Learn-to-Branch**[5] (Guo et al., 2020), **Task-Grouping**[5] (Standley et al., 2020). For general MTL approaches, we compare with following baselines: the **Single-Task** baseline where each task has its own model and is trained independently, popular MTL methods (e.g., **Cross-Stitch** (Misra et al., 2016), **Sluice** (Ruder et al., 2019), **NDDR-CNN** (Gao et al., 2019), **MTAN** (Liu et al., 2019)), and state-of-the-art NAS-based MTL methods (e.g. **DEN** (Ahn et al., 2019), **AdaShare** (Sun et al., 2019), **AutoMTL** (Zhang et al., 2021)).

We use the same backbone model in all baselines and in our approach for fair comparisons. We use Deeplab-ResNet34 (Chen et al., 2017) and MobileNetV2 (Sandler et al., 2018) as the backbone model and the Atrous Spatial Pyramid Pooling (ASPP) architecture as the task-specific head. Both of them are popular architectures for pixel-wise prediction tasks. The branching points of Deeplab-ResNet34 are generated by our branching point detector and then further customized to be five according to He et al. (2016) to reduce search space, each computation block corresponding to one ConvBNReLU block or one Residual Block. Similarly, MobileNetV2 is split into separate Inverted Blocks firstly and then its branching points are defined by merging adjacent blocks into five larger ones with similar computation cost measured by FLOPs.

### 4.2 Performance of Recommended Tree-Structured Multi-Task Models

Table 1∼2 report the real task performance of the recommended tree-structured multi-task models after training using Deeplab-ResNet34. It reports both absolute values of all evaluation metrics and the relative performance. The first column "Model" lists the index of the recommended models. Overall, the recommendation of our framework is consistent with the common belief that MTL can achieve higher task accuracy and improved efficiency for each task by leveraging commonalities across related tasks (Caruana, 1997; Ruder, 2017).

The superiority of our recommender can be observed more clearly in Table 2. With different computation budgets (specified by the number of backbone models in column "Com. Budget"), our recommender could always recommend multi-task architectures with high task performance within the computation constraint. Unlike prior works (Sun et al., 2019; Bruggemann et al., 2020), which have to re-train the whole architecture searching framework when the computational requirement changes, there is no extra effort for our framework to re-predict the top architectures. The recommender can suggest top architectures on the fly by filtering out architectures that do not satisfy the given requirement.

---

[3]We implemented the algorithm ourselves since the work is not open-sourced.

[4]It has implementation on MobileNetV2 only.

[5]Its tree-structured multi-task model for Taskonomy is implemented based on the architecture reported in the paper by ourselves since the work is not open-sourced.

Table 2: Performance of top-1 recommended architectures on Taskonomy using Deeplab-ResNet34 under different computation budgets.

| Models | Com. Budget | FLOPs (%) ↓ | #Params (%) ↓ | Semantic Seg. | | Normal Pred. | | Depth Est. | | Keypoint Det. | | Edge Det. | | Δt ↑ |
|---|---|---|---|---|---|---|---|---|---|---|---|---|---|---|
| | | | | Abs. ↓ | Δt₁ ↑ | Abs. ↑ | Δt₂ ↑ | Abs. ↓ | Δt₃ ↑ | Abs. ↓ | Δt₄ ↑ | Abs. ↓ | Δt₅ ↑ | |
| Ind. Models | - | - | - | 0.5217 | - | 0.8070 | - | 0.0220 | - | 0.2024 | - | 0.2140 | - | - |
| #353 | w/o | -11.31 | -7.90 | 0.5168 | 0.9 | 0.8745 | 8.4 | 0.0195 | 11.4 | 0.2003 | 1.0 | 0.2082 | 2.7 | 4.9 |
| #958 | 4 Models | -22.05 | -21.27 | 0.5268 | -1.0 | 0.8744 | 8.4 | 0.0202 | 8.2 | 0.1887 | 6.8 | 0.2159 | -0.9 | 4.3 |
| #1046 | 3 Models | -41.93 | -41.27 | 0.5368 | -2.9 | 0.8723 | 8.1 | 0.0201 | 8.6 | 0.1987 | 1.8 | 0.2118 | 1.0 | 3.3 |
| #817 | 2 Models | -60.00 | -60.00 | 0.5891 | -12.9 | 0.8725 | 8.1 | 0.0200 | 9.1 | 0.1915 | 5.4 | 0.2105 | 1.6 | 2.3 |
| #0 | 1 Model | -80.00 | -80.00 | 0.5994 | -14.9 | 0.8390 | 4.0 | 0.0265 | -20.5 | 0.1947 | 3.8 | 0.2072 | 3.2 | -4.9 |

## 4.3 Evaluation of the Task Accuracy Estimator

Our recommender can get a **predicted ranking** of all the multi-task architectures based on their estimated task performance from the task accuracy estimator. To evaluate the predicted ranking, we also get an **oracle ranking**, by actual training the multi-task architectures. We use the Pearson correlation coefficient (Pearson's $\gamma$) (Benesty et al., 2009) of the *predicted ranking* and the *oracle ranking* to evaluate the

Table 3: Pearson's $\gamma$ between the predicted ranking and the oracle ranking.

| Method | Deeplab-ResNet34 | | MobileNetV2 | |
|---|---|---|---|---|
| | NYUv2 | Taskonomy | NYUv2 | Taskonomy |
| What-to-Share | -0.478 | -0.147 | -0.4901 | -0.754 |
| Ours | 0.699 | 0.768 | 0.504 / 0.772 | 0.836 |

efficacy of the task accuracy estimator component. We compare with the correlation of **What-to-Share** (Vandenhende et al., 2019), the only existing branched MTL method which could sort the architectures according to task dissimilarity scores. Table 3 reports the correlation results. For reproducibility, the random seed of the experiments is set as 10. For NYUv2 on MobileNetV2, we also conduct the same experiment with seed 20. The range of $\gamma$ is $[-1, 1]$. The larger the value of $\gamma$ is, the stronger the positive correlation, and the better the predicted ranking. Overall, our estimated architecture ranking has a moderately high correlation (i.e., $0.4 \leq \gamma < 0.7$) or even very strong correlation (i.e., $0.7 \leq \gamma < 0.9$) with the oracle ranking according to the interpretation of Pearson's $\gamma$ (Akoglu, 2018), which demonstrates the reliability of the task accuracy estimator and the effectiveness of our recommender. In contrast, What-to-Share produces negative correlations, indicating that their estimations from task dissimilarity are unreliable. Compared with What-to-Share, our recommender improves the correlation significantly.

## 4.4 Comparison with State-of-the-Art MTL Methods

Table 4 summarizes the comparisons with state-of-the-art MTL methods for Taskonomy on Deeplab-ResNet34. Table 5 and 6 report the results on MobileNetV2. Generally, the best multi-task architectures suggested by our recommender could achieve competitive or even higher overall task performance as indicated by the $\Delta t$ columns.

Our work is closest to **What-to-Share** which also ranks all the candidate multi-task architectures and outperforms it by 4.8% in terms of the overall task performance. **Task-Grouping** focuses on deciding how to split the tasks into groups according to the given computation budget so that one group will share the entire backbone model. Compared to **Task-Grouping**, our recommender yields better branching models under the same budget. For instance, when the budget is three models, our top-1 multi-task architecture could achieve higher task performance (3.3% *vs* 2.4%) with lower computation cost (-41.93% *vs* -40%) and number of parameters (-41.27% *vs* -40%) than Task-Grouping.

Compared with manually-design multi-task architectures, **Cross-Stitch**, **Sluice**, **NDDR-CNN**, and **MTAN**, which usually consist of separate networks for each task and define a mechanism for

Table 4: Comparison with state-of-the-art MTL methods for Taskonomy using Deeplab-ResNet34.

| Models | FLOPs (%) ↓ | #Params (%) ↓ | Semantic Seg. Abs. ↓ | $\Delta t_1$ ↑ | Normal Pred. Abs. ↑ | $\Delta t_2$ ↑ | Depth Est. Abs. ↓ | $\Delta t_3$ ↑ | Keypoint Det. Abs. ↓ | $\Delta t_4$ ↑ | Edge Det. Abs. ↓ | $\Delta t_5$ ↑ | $\Delta t$ ↑ |
|---|---|---|---|---|---|---|---|---|---|---|---|---|---|
| Ind. Models | - | - | 0.5217 | - | 0.807 | - | 0.022 | - | 0.2024 | - | 0.214 | - | - |
| What-to-Share | -0.13 | -0.01 | 0.5378 | -3.1 | 0.8696 | 7.8 | 0.0233 | -5.9 | 0.2019 | 0.2 | 0.2113 | 1.3 | 0.1 |
| Task-Grouping | -40.00 | -40.00 | 0.5388 | -3.3 | 0.8743 | 8.3 | 0.0202 | 8.2 | 0.2037 | -0.6 | 0.2151 | -0.5 | 2.4 |
| Cross-Stitch | 0.00 | 0.00 | 0.57 | -9.3 | 0.779 | -3.5 | 0.021 | 4.5 | 0.199 | 1.7 | 0.217 | -1.4 | -1.6 |
| Sluice | 0.00 | 0.00 | 0.596 | -14.2 | 0.795 | -1.5 | 0.023 | -4.5 | 0.196 | 3.2 | 0.207 | 3.3 | -2.8 |
| NDDR-CNN | 8.38 | 8.20 | 0.599 | -14.8 | 0.8 | -0.9 | 0.022 | 0.0 | 0.196 | 3.2 | 0.203 | 5.1 | -1.5 |
| MTAN | -10.55 | -9.80 | 0.621 | -19.0 | 0.787 | -2.5 | 0.022 | 0.0 | 0.197 | 2.7 | 0.206 | 3.7 | -3.0 |
| Learn-to-Branch | **-68.11** | **-67.67** | 0.5214 | 0.1 | 0.8503 | 5.4 | 0.0235 | -6.8 | 0.2021 | 0.1 | 0.2171 | -1.4 | -0.5 |
| DEN | 2.15 | -77.60 | 0.737 | -41.3 | 0.786 | -2.6 | 0.026 | -18.2 | 0.192 | 5.1 | 0.203 | 5.1 | -10.4 |
| AdaShare | -5.42 | -71.20 | 0.562 | -7.7 | 0.802 | -0.6 | 0.022 | 0.0 | 0.191 | **5.6** | 0.200 | 6.5 | 0.8 |
| AutoMTL | -3.85 | -50.10 | 0.536 | -2.7 | 0.873 | 8.2 | 0.021 | 4.5 | 0.191 | **5.6** | 0.197 | **7.9** | 4.7 |
| **Top-1 w/o budget** | -11.31 | -7.90 | 0.5168 | **0.9** | 0.8745 | **8.4** | 0.0195 | **11.4** | 0.2003 | 1.0 | 0.2082 | 2.7 | **4.9** |
| **Top-1 within 3 models** | -41.93 | -41.27 | 0.5368 | -2.9 | 0.8723 | 8.1 | 0.0201 | 8.6 | 0.1987 | 1.8 | 0.2118 | 1.0 | 3.3 |

Table 5: Comparison with Branched MTL methods for NYUv2 using MobileNetV2.

| Model | FLOPs (%) ↓ | #Params (%) ↓ | Semantic Seg. mIoU ↑ | Pixel Acc. ↑ | $\Delta t_1$ ↑ | Surface Normal Prediction Error ↓ Mean | Median | $\theta$, within ↑ 11.25° | 22.5° | 30° | $\Delta t_2$ ↑ | Depth Estimation Error ↓ Abs. | Rel. | $\delta$, within ↑ 1.25 | $1.25^2$ | $1.25^3$ | $\Delta t_3$ ↑ | $\Delta t$ ↑ |
|---|---|---|---|---|---|---|---|---|---|---|---|---|---|---|---|---|---|---|
| Ind. Models | - | - | 20.36 | **49.44** | - | 18.17 | 16.62 | 28.37 | 70.20 | 85.58 | - | 0.77 | 0.28 | 47.92 | 78.46 | 92.81 | - | - |
| What-to-Share | -8.41 | -0.30 | **21.10** | 49.03 | **1.4** | 17.82 | **15.71** | 30.61 | 72.60 | 86.08 | 3.9 | 0.67 | 0.25 | 54.71 | 83.25 | 94.97 | 9.3 | 4.8 |
| BMTAS | -64.46 | -33.41 | 18.98 | 48.40 | -4.4 | **17.71** | 16.09 | 29.74 | **72.70** | **86.90** | 3.1 | **0.60** | 0.24 | **60.73** | **87.25** | 96.33 | **15.9** | 4.9 |
| **Top-1** | **-66.67** | **-66.67** | 19.36 | 48.97 | -2.9 | 17.99 | 16.02 | **31.43** | 70.41 | 84.65 | 2.9 | 0.61 | **0.23** | 60.02 | 86.89 | **96.34** | 15.7 | **5.2** |

feature sharing between independent networks, our recommended architectures perform higher task performance (4.9% *vs* -1.6%/-2.8%/-1.5%/-3.0%) with computation cost (-11.31% *vs* 0%/8.38%/-10.55%) and the number of parameters reduction (-7.90% *vs* 0%/8.20%/-9.80%).

We also compare with NAS-based methods, including NAS-based branched MTL methods such as **Learn-to-Branch** and **BMTAS**, and NAS-based general MTL approaches such as **DEN**, **AdaShare**, and **AutoMTL**. Learn-to-Branch and BMTAS explore the same tree-structured architecture design space as our recommender. However, since they resort to integrating space searching with network training, the searched multi-task models are usually sub-optimal. Instead, our recommender could overcome the limitation to identify multi-task architectures with higher task performance, 5.4% higher than Learn-to-Branch, and 0.3%/4.1% higher than BMTAS on NYUv2 and Taskonomy using MobileNetV2 as shown in Table 5 and 6. When comparing to DEN, AdaShare, and AutoMTL, our recommender identifies multi-task architectures with competitive task performance (4.9% *vs* -10.4%/0.8%/4.7%), even though the search space of those methods are larger and more complex than our tree-structured multi-task model space.

Table 6: Comparison with Branched MTL methods for Taskonomy using MobileNetV2.

| Models | FLOPs (%) ↓ | #Params (%) ↓ | Semantic Seg. Abs. ↓ | $\Delta t_1$ ↑ | Normal Pred. Abs. ↑ | $\Delta t_2$ ↑ | Depth Est. Abs. ↓ | $\Delta t_3$ ↑ | Keypoint Det. Abs. ↓ | $\Delta t_4$ ↑ | Edge Det. Abs. ↓ | $\Delta t_5$ ↑ | $\Delta t$ ↑ |
|---|---|---|---|---|---|---|---|---|---|---|---|---|---|
| Ind. Models | - | - | 0.5217 | - | 0.807 | - | 0.022 | - | 0.2024 | - | 0.214 | - | - |
| What-to-Share | -5.02 | -0.18 | 1.0283 | -1.9 | 0.7656 | -0.1 | 0.0275 | **0.7** | 0.2417 | -0.9 | 0.2688 | -0.3 | -0.5 |
| BMTAS | **-78.47** | **-76.32** | 1.0239 | -1.4 | 0.7511 | -2.0 | 0.0322 | -16.2 | 0.2202 | 8.1 | 0.2508 | 6.5 | -1.0 |
| Task-Grouping | -25.11 | -5.21 | 0.9965 | 1.3 | 0.7678 | 0.2 | 0.0287 | -3.6 | 0.2323 | 3.0 | 0.2591 | 3.4 | 0.9 |
| **Top-1** | -53.99 | -44.86 | 0.977 | **3.2** | 0.7625 | -0.5 | 0.0277 | 0.0 | 0.2232 | 6.8 | 0.2519 | 6.0 | **3.1** |

## 5 Conclusion

This paper proposes a tree-structured multi-task model recommender that predicts the top-$k$ architectures with high task performance given a set of tasks, an arbitrary CNN backbone model, and a user-specified computation budget. Our recommender consists of three key components, a branching point detector that automatically detects branching points in any given CNN backbone model, a design space enumerator that enumerates all the multi-task architecture in the design space, and a task accuracy estimator that predicts the task performance of multi-task architectures without performing actual training. Experiments on popular MTL benchmarks demonstrate the superiority and reliability of our recommender compared with state-of-the-art approaches.

**Limitations and Broader Impact Statement**. Our research facilitates the adoption of deep learning techniques to solve many tasks at once in resource-constraint scenarios. It also promotes the leverage of multi-task learning to increase task performance and computation efficiency. It has a positive impact on applications that tackle multiple tasks such as environment perceptions for autonomous vehicles and human-computer interactions in robotic, mobile, and IoT applications. The negative social impact of our research is difficult to predict since it shares the same pitfalls with general deep learning techniques that suffer from dataset bias, adversarial attacks, fairness, etc.

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
