# A Tree-Structured Multi-Task Model Recommender

**Anonymous**[1]

[1]Anonymous Institution

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

|---|---|---|---|---|---|---|---|---|---|---|---|---|---|---|---|---|---|---|
| | | | mIoU ↑ | Pixel Acc. ↑ | $\Delta t_1$ ↑ | Error ↓ | | $\theta$, within ↑ | | | $\Delta t_2$ ↑ | Error ↓ | | $\delta$, within ↑ | | | $\Delta t_3$ ↑ | |
| | | | | | | Mean | Median | 11.25° | 22.5° | 30° | | Abs. | Rel. | 1.25 | $1.25^2$ | $1.25^3$ | | |
| Ind. Models | - | - | **26.50** | **58.20** | - | 17.70 | 16.30 | 29.40 | 72.30 | **87.30** | - | 0.62 | 0.24 | 57.80 | 85.80 | 96.00 | - | - |
| #0 | -66.67 | -66.66 | 25.23 | 57.69 | -2.8 | **17.14** | 15.15 | 35.85 | 72.20 | 85.54 | **6.0** | **0.55** | 0.23 | 63.85 | 89.38 | 97.03 | 5.8 | **3.0** |
| #45 | -36.56 | -35.45 | 25.18 | 57.36 | -3.2 | 17.26 | **14.93** | **36.33** | 72.27 | 85.16 | 6.4 | 0.58 | **0.22** | 62.70 | 88.79 | 96.93 | 5.5 | 2.9 |
| #50 | -46.87 | -46.13 | 24.72 | 56.71 | -4.6 | 17.24 | 15.13 | 32.17 | **72.66** | 85.75 | 3.6 | 0.56 | 0.23 | 63.87 | 88.72 | 96.81 | 5.6 | 1.5 |
| #37 | -34.87 | -33.70 | 26.30 | 57.94 | **-0.6** | 17.24 | 15.16 | 35.78 | 71.90 | 85.43 | 5.7 | 0.61 | **0.22** | 60.09 | 87.18 | 96.31 | 2.6 | 2.6 |
| #49 | -46.87 | -46.13 | 25.56 | 57.62 | -2.3 | 17.77 | 15.70 | 33.18 | 70.99 | 84.64 | 2.3 | **0.55** | **0.22** | **64.62** | **89.78** | **97.55** | 7.8 | 2.6 |

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

# A  Associated Two-Task Models Identifier

As introduced in Section 3.1, to estimate the task accuracy for a multi-task model with three or more tasks, we first need to identify its associated two-task models. To achieve the goal, we propose an iterative algorithm. The pseudocode of the algorithm is shown in Algorithm 1. The layout representation proposed in Section 3.2 is used to refer to the multi-task models for simplicity.

---

**Algorithm 1** Associated Two-Task Layout Identifier

---

**Input:** A multi-task layout $\mathbf{L}$
**Output:** A dictionary $D$ containing associated two-task layouts
1: $D \leftarrow \{\}$
2:
3: **for** $t_1 \in \mathbf{L}$ **do**  ▷ For each task in $\mathbf{L}$
4:     $D[t_1] \leftarrow []$  ▷ Store the associated two-task layouts for $t_1$
5:
6:     **for** $t_2 \in \mathbf{L}$ **do**  ▷ Identify for all possible task combinations
7:         ▷ Skip identical task
8:         **if** $t_1 = t_2$ **then**
9:             **continue**
10:         **end if**
11:
12:         ▷ Initiate the branching out point to the maximum possible branching point
13:         $b = B + 1$
14:         **for** $i = 1 \rightarrow B$ **do**  ▷ Check every possible branching point
15:             $flag \leftarrow$ **false**  ▷ Store if $t_1$ and $t_2$ share at the $i$-th branching point
16:             **for** $L_i^j \in \mathbf{L}$ **do**  ▷ Check for each task set at the $i$-th branching point
17:                 **if** $t_1 \in L_i^j$ and $t_2 \in L_i^j$ **then**
18:                     $flag \leftarrow$ **true**  ▷ There exists a task set has both $t_1$ and $t_2$
19:                     **break**
20:                 **end if**
21:             **end for**
22:             **if** $flag$ is false **then**  ▷ $t_1$ and $t_2$ don't share at the $i$-th branching point
23:                 $b = i$
24:                 **break**
25:             **end if**
26:         **end for**
27:
28:         ▷ Store the associated two-task layouts for $t_1$ and $t_2$ based on $b$
29:         $\mathbf{L}' \leftarrow [[\{t_1, t_2\}]_1, \ldots, [\{t_1, t_2\}]_{b-1}, [\{t_1\}, \{t_2\}]_b, \ldots, [\{t_1\}, \{t_2\}]_B]$ ▷ $t_1$ and $t_2$ separate at $b$-th branching point
30:         $D[t_1]$.append($\mathbf{L}'$)
31:     **end for**
32: **end for**

---

# B  Property of Layout

This section introduces the properties of the proposed layout definitions. Given a layout $\mathbf{L} = [L_1, \ldots, L_i, L_{i+1}, \ldots, L_B]$, for $L_{i+1} = [L_{i+1}^1, L_{i+1}^2, \cdots]$, we have,

$$\forall L_i^q \quad \exists \{L_{i+1}^{a_1}, L_{i+1}^{a_2}, \cdots\} : L_i^q = \bigcup L_{i+1}^{a_p}$$
$$\forall L_{i+1}^p \quad \exists L_i^q : L_{i+1}^p \subseteq L_i^q$$

In other words, these are two properties for successive lists of task sets in a layout. 490

- Any task set in the $i$-th list of task sets is the union of some task sets in the $i + 1$-th list. 491
- Any task set in the $i + 1$-th list of task sets is a subset of one task set in the $i$-th list. 492

The above two properties can be derived directly from the fact that a layout describes a tree. 493

## C  Complete Definition of Layout Cut 494

We first define available branching points of a layout as follows. 495

**Definition C.1** (Available Branching Point). *Given a layout, a branching point is available if the list* 496
*of task sets in the branching point is the same as that in all the subsequent branching points, and there* 497
*exists a task set that contains at least 2 tasks.* 498

From the definition, we could identify available branching points in a given layout by comparing 499
the list of task sets from the $B$-th branching point up to its previous ones until reaching a different 500
list, then checking whether there exists an eligible task set with at least 2 elements. As shown in 501
Figure 3, the available branching points of the left layout are the last three branching points. 502
Then the complete definition of *Layout Cut* for enumerating all possible layouts is defined as 503
follows. 504

**Definition C.2** (Layout Cut). *A layout cut is an operator that transforms one layout to another layout* 505
*by selecting a task set containing at least two tasks from an available branching point and dividing it* 506
*into two task sets.* 507

*Formally, given a layout* $\mathbf{L}$*, the format of a cut* $\mathbf{C}$ *would be like,* 508

$$\mathbf{C} = \{i, L_i^j, [L_{i\,1}^j, L_{i\,2}^j]\}(|L_i^j| \geq 2)$$

*where* $L_i^j$ *is the selected task set containing at least 2 tasks from the i-th branching point (available) of* 509
$\mathbf{L}$ *to be divided into 2 sub task sets* $L_{i\,1}^j$ *and* $L_{i\,2}^j$*.* 510

*When applying the cut* $\mathbf{C}$ *on the layout* $\mathbf{L}$*, the selected task set* $L_i^j$ *will be divided into* $L_{i\,1}^j$ *and* $L_{i\,2}^j$ 511
*in the new layout* $\mathbf{L}'$*.* 512

*Notice that the i-th branching point must be an available branching point of* $\mathbf{L}$*. Then according* 513
*to Definition C.1, we know that the list of task sets in the i-th branching point and its subsequent* 514
*branching points are the same. In other words, the selected task set* $L_i^j$ *also exists in the subsequent* 515
*branching points. Therefore we define that when applying the cut* $\mathbf{C}$ *on the layout* $\mathbf{L}$*, all the task sets in* 516
*the subsequent branching points that are the same as* $L_i^j$ *will be divided into* $L_{i\,1}^j$ *and* $L_{i\,2}^j$ *in the new* 517
*layout* $\mathbf{L}'$ *as well.* 518

For example in Figure 3, there are two different cuts applied on the left given layout which has 519
4 tasks and 5 branching points. The first cut is $\mathbf{C} = \{3, (3, 4), [(3), (4)]\}$, which divides the task set 520
$(3, 4)$ into task sets $(3)$ and $(4)$ at the third branching point as well as all its subsequent branching 521
points. Similarly, the second one, $\mathbf{C} = \{4, (1, 2), [(1), (2)]\}$, divides the task set $(1, 2)$ into $(1)$ and 522
$(2)$ at the fourth and fifth branching points. The effects of the defined cuts are just like the red and 523
orange dashed lines applied on the left layout and the generated new layouts are illustrated on the 524
right side. It's worth mentioning that the available branching points in the new layout $\mathbf{L}'$ may need 525
to be updated as the second example in Figure 3. Specifically, the available branching points of the 526
generated layout $\mathbf{L}'$ would be all the follow-up branching points of the selected branching point of 527
the cut $\mathbf{C}$ and the selected one itself. 528

**Corollary C.2.1.** *The maximum number of cuts that can be applied to the initial layout* $\mathbf{L}_0$ *is* $T - 1$*,* 529
*where* $T$ *is the number of tasks.* 530

[Figure]

Figure 3: Examples of applying a cut on a given layout.

*Proof of Corollary C.2.1.* The corollary can be proved by contradiction easily.

If we apply $T$ cuts on the initial layout $\mathbf{L}_0$, $[\{t_1, \ldots, t_T\}]$ at the $B$-th branching point will be divided into $T+1$ task sets, since the task sets in the subsequent branching points are also influenced when applying a cut according to Definition C.2, which means the task sets in $B$-th branching point will be divided into subsets by all the $T$ cuts. However, it is impossible that $[\{t_1, \ldots, t_T\}]$ is divided into $T+1$ task sets since there are only $T$ tasks in it. □

## D  Layouts Enumerator

As introduced in Section 3.2, we propose a cut-based algorithm to enumerate all possible layouts in the design space fully in a recursive way.

---

**Algorithm 2** Layouts Enumerator

---

**Input**: $T$ tasks and a backbone model with $B$ branching points
**Output**: A set of all possible layouts $S$
 1: **function** ENUMERATOR(**L**)
 2:     $S \leftarrow set()$
 3:     ▷ Exit Case: The number of cuts applied to **L** is $T-1$
 4:     **if** **L**.*num_cut* $= T-1$ **then**
 5:         **return** $S$
 6:     **end if**
 7:
 8:     ▷ Enumerate all possible layout cuts on **L**
 9:     **for** $i \in$ **L**.*avail_bp* **do**                          ▷ Cut for every available branching points
10:         **for** $L_i^j \in$ **L do**                          ▷ Cut for each task set at the $i$-th branching point
11:             **if** $|L_i^j| = 1$ **then**          ▷ If the selected task set has only 1 task, no more cut applied
12:                 **continue**
13:             **end if**
14:             **for** $[L_{i\,1}^j, L_{i\,2}^j] \in$ partition$(L_i^j)$ **do**                          ▷ For every possible partition of $L_i^j$
15:                 $\mathbf{C} \leftarrow \{i, L_i^j, [L_{i\,1}^j, L_{i\,2}^j]\}$
16:                 $\mathbf{L}' \leftarrow$ apply_cut(**L**, **C**)
17:                 $\mathbf{L}'$.*num_cut*$+ = 1$                          ▷ Update the number of cuts applied
18:                 $\mathbf{L}'$.*avail_bp* $\leftarrow [i, \cdots, B]$                          ▷ Update the available branching points
19:                 $S$.append(**L**')
20:                 $S' \leftarrow$ ENUMERATOR(**L**')                          ▷ Enumerate cuts for **L**' recursively
21:                 $S+ = S'$
22:             **end for**

23:        **end for**          565

24:     **end for**          566

25:     **return** $S$          567

26:  **end function**          568

27:          569

28:  $\mathbf{L}_0 \leftarrow \underbrace{[[\{t_1, \ldots, t_T\}], \cdots, [\{t_1, \ldots, t_T\}]]}_{B}$          570

29:  $\mathbf{L}_0.num\_cut \leftarrow 0$          571

30:  $\mathbf{L}_0.avail\_bp \leftarrow [1, \cdots, B]$          572

31:  $S \leftarrow \textsc{Enumerator}(\mathbf{L}_0)$          573

---

Considering the recursive tree of the enumerator, the time complexity of this algorithm is $O\left([(2^{T-1} - 1) \cdot B]^{T-1}\right)$, where $T - 1$ is the depth of the recursive tree according to Corollary C.2.1 and $O((2^{T-1} - 1) \cdot B)$ is the branching factor of it. Here $O(2^{T-1} - 1)$ is the time complexity of partitioning a set into two subsets (line 14).

## E  Proof of Theorem 3.1

*Proof of Theorem 3.1.* We want to show that for $T$ tasks and a backbone model with $B$ branching points, the layouts enumerated by the cut-based algorithm can cover all tree-structured multi-task architectures. We can prove it by induction.

**Base Case**: When $B = 1$, all possible tree-structured multi-task architectures can be considered as dividing the $T$ tasks into $n$ groups $\{\{\Gamma_1, \Gamma_2, \cdots, \Gamma_n\} \mid \bigcup_{i=1}^{n} \Gamma_i = \mathcal{T}, n \leq T\}$ where $\Gamma_i$ is a task set, indicating the sharing pattern across tasks at the only branching point. Then each tree-structured multi-task model can be represented as a layout $\mathbf{L} = [[\Gamma_1, \Gamma_2, \cdots, \Gamma_n]]$ and such a layout can be generated by applying $n - 1$ cuts on the initial layout $\mathbf{L}_0 = [[\mathcal{T}]]$ inductively.

The first cut is $\mathbf{C}_1 = \{1, \mathcal{T}, [\Gamma_1, \bigcup_{i=2}^{n} \Gamma_i]\}$, which divides the initial task set $\mathbf{T}$ into $\Gamma_1$ and $\bigcup_{i=2}^{n} \Gamma_i$. By applying $\mathbf{C}_1$ on $\mathbf{L}_0$, we can get $\mathbf{L}_1 = [[\Gamma_1, \bigcup_{i=2}^{n} \Gamma_i]]$. Then the second cut is $\mathbf{C}_2 = \{1, \bigcup_{i=2}^{n} \Gamma_i, [\Gamma_2, \bigcup_{i=3}^{n} \Gamma_i]\}$, and we can get $\mathbf{L}_2 = [[\Gamma_1, \Gamma_2, \bigcup_{i=3}^{n} \Gamma_i]]$. Similarly the $k$-th cut is $\mathbf{C}_k = \{1, \bigcup_{i=k}^{n} \Gamma_i, [\Gamma_k, \bigcup_{i=k+1}^{n} \Gamma_i]\}$, and we can get $\mathbf{L}_k = [[\Gamma_1, \Gamma_2, \cdots, \Gamma_k, \bigcup_{i=k+1}^{n} \Gamma_i]]$. Finally, the $(n-1)$-th cut is $\mathbf{C}_{n-1} = \{1, \Gamma_{n-1} \bigcup \Gamma_n, [\Gamma_{n-1}, \Gamma_n]\}$, and we can get $\mathbf{L}_{n-1} = [[\Gamma_1, \Gamma_2, \cdots, \Gamma_{n-1}, \Gamma_n]] = \mathbf{L}$, which is any layout we want. The overall chain would be like,

$$\mathbf{L}_0 = [[\mathcal{T}]] \xrightarrow{\mathbf{C}_1 = \{1, \mathcal{T}, [\Gamma_1, \bigcup_{i=2}^{n} \Gamma_i]\}} \mathbf{L}_1 = [[\Gamma_1, \bigcup_{i=2}^{n} \Gamma_i]]$$

$$\vdots$$

$$\xrightarrow{\mathbf{C}_k = \{1, \bigcup_{i=k}^{n} \Gamma_i, [\Gamma_k, \bigcup_{i=k+1}^{n} \Gamma_i]\}} \mathbf{L}_k = [[\Gamma_1, \Gamma_2, \cdots, \Gamma_k, \bigcup_{i=k+1}^{n} \Gamma_i]]$$

$$\vdots$$

$$\xrightarrow{\mathbf{C}_{n-1} = \{1, \Gamma_{n-1} \bigcup \Gamma_n, [\Gamma_{n-1}, \Gamma_n]\}} \mathbf{L}_{n-1} = [[\Gamma_1, \Gamma_2, \cdots, \Gamma_{n-1}, \Gamma_n]] = \mathbf{L}$$

In summary, when $B = 1$, any tree-structured multi-task model can be generated by the cut-based layout enumeration algorithm through the cuts chain above.

**Inductive Hypothesis**: Any tree-structured multi-task model for $T$ tasks based on a backbone model with $B$ branching points can be enumerated through the cut-based layout enumeration

algorithm completely. 

**Inductive Goal**: The completeness also holds for any backbone model with $B+1$ branching point.

**Inductive Steps**: Suppose we have a tree-structured multi-task model $M$ built on a backbone model with $B$ branching point whose layout representation is $\mathbf{L} = [L_1, L_2, \cdots, L_B]$ where $L_i$ is a list of task sets. We can derive a new layout $\mathbf{L}' = [L_1, L_2, \cdots, L_B, L_{B+1}]$ with $B+1$ branching point from $\mathbf{L}$ easily by keeping the first $B$ levels unchanged and adding one more level $L_{B+1}$, which is equivalent to adding a new level with leaf nodes to $M$. There are two cases about this new level:

(1) $L_{B+1} = L_B$. Since according to Definition C.2, a cut will influence all the subsequent branching points from the selected one, the cuts we used to generate the layout $\mathbf{L}$ from the initial layout $\mathbf{L}_0$ can generate the new layout $\mathbf{L}'$ as well.

(2) $L_{B+1} \neq L_B$. Suppose we have,

$$L_B = [\Gamma_1, \cdots, \Gamma_n]$$
$$L_{B+1} = [\Gamma'_1, \cdots, \Gamma'_m]$$

Then according to the two properties of a layout introduced in Section B, we have,

$$\forall \Gamma_i \exists \{\Gamma'_{a_1}, \Gamma'_{a_2}, \cdots, \Gamma'_{a_k}\} : \Gamma_i = \bigcup_{j=1}^{k} \Gamma'_{a_j}$$

For any $\Gamma_i$ with such a set $|\{\Gamma'_{a_1}, \Gamma'_{a_2}, \cdots, \Gamma'_{a_k}\}| > 1$, we could apply $k-1$ cuts whose selected branching points are $B+1$ on the layout $\hat{\mathbf{L}}$ which is derived from the layout $\mathbf{L}$ with one more level $L_{B+1} = L_B$ to divide $\Gamma'_i = \Gamma_i$ at the $B+1$ level into $k$ task sets $\Gamma'_{a_1}, \Gamma'_{a_2}, \cdots, \Gamma'_{a_k}$. This process is the same as the process that we divide the task set $\mathcal{T}$ into $n$ groups in the base case. Therefore if we have $r$ such $\Gamma_i$ at the $B$-th level in the target layout $\mathbf{L}'$, we can apply $r(k-1)$ cuts on the layout $\hat{\mathbf{L}}$ inductively to generate $\mathbf{L}'$.

In summary, if any tree-structured multi-task model for $T$ tasks based on a backbone model with $B$ branching point can be enumerated through the cut-based layout enumeration algorithm completely, we can achieve it for any backbone model with $B+1$ branching point as well. Together with the base case, we have proved that the search space of tree-structured multi-task models can be explored by the cut-based layout enumeration algorithm completely. □

## F  Branching Point Detector

As in described in Section 3.3, we propose a 2-stage branching point detector to partition the user-provided backbone into sequential blocks.

An example is illustrated in Figure 4. In the first stage, if the size of the minimum cut of the computation graph is 1 as in Figure 4(a), the graph can be divided into two subgraphs with only one link between them according to the definition of the minimum cut. Then for each subgraph, the same division occurs if it has a minimum cut of size 1 (e.g., blocks A and B). However, if the size of the minimum cut in the subgraph is greater than 1 (e.g., block C) or the subgraph has only one node (e.g., blocks D and E), the subgraph can no longer be partitioned. After the first stage, each candidate block is assigned a label to reveal its property for merging. If a block contains only unparameterized layers and normalization layers, it is labeled as $U$, otherwise it is labeled as $P$. Then the merging carries out from bottom to the top. If the label of a block is $U$, it should be merged with its next blocks until reaching a block labeled $P$, otherwise no action is required.

The whole process is illustrated in the following pseudocode.

[Figure]

| (a) Stage 1: Divisor | (b) Stage 2: Merger |

Figure 4: Process of the branching point detector.

**Algorithm 3** Branching Point Detector

**Input:** A backbone model $M$
**Output:** A set of subgraphs $S'$ indicating sequential blocks
1: **function** DIVISOR($G = (V, E)$)
2:      $S \leftarrow set()$
3:      ▷ Case 1: The graph has only 1 node
4:      **if** $|G.V| = 1$ **then**
5:         $S$.append($G$)
6:         **return** $S$
7:      **end if**
8:
9:      $C \leftarrow min\_cut(G)$                 ▷ Find the minimum cut $C = (A, B)$ of $G$
10:      ▷ Case 2: The size of the minimum cut is greater than 1
11:      **if** $|C.E| > 1$ **then**              ▷ $C.E = \{(u, v) \in G.E | u \in A, v \in B\}$
12:         $S$.append($G$)
13:      **else**
14:         ▷ Case 3: The size of the minimum cut is 1
15:         $S_A \leftarrow$DIVISOR($G_A = (A, E_A)$)        ▷ $E_A = \{(u, v) \in G.E | u \in A, v \in A\}$
16:         $S_B \leftarrow$DIVISOR($G_B = (B, E_B)$)        ▷ $E_B = \{(u, v) \in G.E | u \in B, v \in B\}$
17:         $S$.append($S_A, S_B$)
18:      **end if**
19:      **return** $S$
20: **end function**
21:
22: **function** MERGER($S$)
23:      $S$.reverse()              ▷ Handle merging from bottom to the top
24:      **for** $B \in S$ **do**
25:         **if** $B$.label **is** $P$ **then**
26:             **continue**
27:         **else**        ▷ $B$ is a block with only unparameterized layers and normalization layers
28:             $S[B] \leftarrow B + S[B + 1]$             ▷ Merge $B$ with $B + 1$
29:             **if** $S[B + 1]$.label **is** $P$ **then** ▷ Update the label of new $B$ according to the label of $B + 1$
30:                 $S[B]$.label $\leftarrow P$
31:             **else**
32:                 $S[B]$.label $\leftarrow U$

```
33:            end if                                                              673
34:          end if                                                                674
35:        end for                                                                 675
36:   end function                                                                 676
37:                                                                                677
38:   G ← CG(M)                        ▷ Convert the model to its computation graph (CG)   678
39:   S ← Divisor(G)                        ▷ Stage 1: Divide the model into candidate blocks   679
40:   S′ ← Merger(S)             ▷ Stage 2: Merge auxiliary blocks with adjacent primary blocks   680
```

## G  Hyper-Parameters Settings

Table 5 summarizes the hyper-parameters used in 2-task and multi-task model training. The settings are chosen by experience in previous works (Sun et al., 2019; Zhang et al., 2021).

Table 5: Hyper-parameters for training NYUv2, and Tiny-Taskonomy.

| Dataset | lr | lr decay | epoch |
|---|---|---|---|
| NYUv2 | 0.001 | 0.5/4,000 iters | 20,000 |
| Tiny-Taskonomy | 0.0001 | 0.3/10,000 iters | 50,000 |

## H  More Recommended Multi-Task Models

Table 6 and 7 report more recommended multi-task models on MobileNetV2. Table 8 is a detailed version of Table 2 to include the top-5 recommended models within different computation budgets.

Table 6: Performance of top-5 recommended models for NYUv2 using MobileNetV2.

| Model | FLOPs (%) ↓ | #Params (%) ↓ | Semantic Seg. | | | Surface Normal Prediction | | | | | | Depth Estimation | | | | | Δt ↑ |
|---|---|---|---|---|---|---|---|---|---|---|---|---|---|---|---|---|---|
| | | | mIoU ↑ | Pixel Acc. ↑ | Δt₁ ↑ | Error ↓ | | θ, within ↑ | | | Δt₂ ↑ | Error ↓ | δ, within ↑ | | | Δt₃ ↑ | |
| | | | | | | Mean | Median | 11.25° | 22.5° | 30° | | Abs. Rel. | 1.25 | 1.25² | 1.25³ | | |
| Ind. Models | - | - | 20.36 | 49.44 | - | 18.17 | 16.62 | 28.37 | 70.20 | 85.58 | - | 0.77 | 0.28 | 47.92 | 78.46 | 92.81 | - | - |
| #0 | -66.67 | -66.67 | 19.36 | 48.97 | -2.9 | **17.99** | 16.02 | **31.43** | 70.41 | 84.65 | **2.9** | 0.61 | **0.23** | 60.02 | **86.89** | **96.34** | **15.7** | **5.2** |
| #7 | -33.16 | -33.41 | 19.60 | 48.33 | -3.0 | 18.00 | **15.96** | 30.39 | 71.15 | 85.22 | 2.6 | 0.61 | **0.23** | 60.03 | 86.78 | 96.31 | 15.7 | 5.1 |
| #11 | -30.83 | -27.12 | 20.04 | 48.82 | -1.4 | 18.21 | 16.38 | 30.37 | 68.95 | 84.45 | 1.0 | **0.60** | **0.23** | **60.37** | 84.04 | 96.16 | 15.3 | 5.0 |
| #10 | -22.89 | -8.24 | 19.42 | 48.49 | -3.3 | 18.06 | 16.07 | 29.48 | 71.12 | 85.26 | 1.8 | 0.63 | 0.24 | 58.23 | 85.57 | 95.45 | 13.2 | 3.9 |
| #9 | -16.35 | -3.60 | **21.09** | **50.25** | **2.6** | 18.10 | 16.73 | 26.85 | **72.13** | **87.11** | -0.2 | 0.64 | 0.25 | 57.10 | 85.07 | 95.32 | 11.6 | 4.7 |

Table 7: Performance of top-5 recommended models for Taskonomy using MobileNetV2.

| Models | FLOPs (%) ↓ | #Params (%) ↓ | Semantic Seg. | | Normal Pred. | | Depth Est. | | Keypoint Det. | | Edge Det. | | Δt ↑ |
|---|---|---|---|---|---|---|---|---|---|---|---|---|---|
| | | | Abs. ↓ | Δt₁ ↑ | Abs. ↑ | Δt₂ ↑ | Abs. ↓ | Δt₃ ↑ | Abs. ↓ | Δt₄ ↑ | Abs. ↓ | Δt₅ ↑ | |
| Ind. Models | - | - | 1.0096 | - | 0.7662 | - | 0.0277 | - | 0.2395 | - | 0.2681 | - | - |
| #3221 | -53.99 | -44.86 | 0.9770 | 3.2 | 0.7625 | -0.5 | 0.0277 | 0.0 | 0.2232 | 6.8 | 0.2519 | 6.0 | 3.1 |
| #3220 | -52.73 | -41.17 | 1.0120 | -0.2 | 0.7624 | -0.5 | 0.0273 | 1.4 | 0.2269 | 5.3 | 0.2443 | 8.9 | 3.0 |
| #2947 | -60.26 | -60.05 | 1.0179 | -0.8 | 0.7510 | -2.0 | 0.0275 | 0.7 | 0.2117 | 11.6 | 0.2547 | 5.0 | 2.9 |
| #3215 | -53.99 | -44.86 | 1.0066 | 0.3 | 0.7620 | -0.5 | 0.0274 | 1.1 | 0.2250 | 6.1 | 0.2552 | 4.8 | 2.3 |
| #3261 | -59.01 | -56.27 | 1.0359 | -2.6 | 0.7489 | -2.3 | 0.0273 | 1.4 | 0.2060 | 14.0 | 0.2477 | 7.6 | **3.6** |

Table 8: Performance of top-5 recommended models for Taskonomy using Deeplab-ResNet34 under different computation budgets.

| Models | FLOPs (%) ↓ | #Params (%) ↓ | Semantic Seg. | | Normal Pred. | | Depth Est. | | Keypoint Det. | | Edge Det. | | $\Delta t$ ↑ |
|---|---|---|---|---|---|---|---|---|---|---|---|---|---|
| | | | Abs. ↓ | $\Delta t_1$ ↑ | Abs. ↑ | $\Delta t_2$ ↑ | Abs. ↓ | $\Delta t_3$ ↑ | Abs. ↓ | $\Delta t_4$ ↑ | Abs. ↓ | $\Delta t_5$ ↑ | |
| Ind. Models | - | - | 0.5217 | - | 0.8070 | - | 0.0220 | - | 0.2024 | - | 0.2140 | - | - |
| No computation budget | | | | | | | | | | | | | |
| #353 | -11.31 | -7.90 | 0.5168 | 0.9 | 0.8745 | 8.4 | 0.0195 | 11.4 | 0.2003 | 1.0 | 0.2082 | 2.7 | **4.9** |
| #352 | -9.43 | -1.49 | 0.5166 | 1.0 | 0.8741 | 8.3 | 0.0200 | 9.1 | 0.1992 | 1.6 | 0.2116 | 1.1 | 4.2 |
| #958 | -22.05 | -21.27 | 0.5268 | -1.0 | 0.8744 | 8.4 | 0.0202 | 8.2 | 0.1887 | 6.8 | 0.2159 | -0.9 | 4.3 |
| #480 | -22.83 | -21.48 | 0.5168 | 0.9 | 0.8734 | 8.2 | 0.0210 | 4.5 | 0.2018 | 0.3 | 0.2146 | -0.3 | 2.7 |
| #360 | -22.05 | -21.27 | 0.5178 | 0.7 | 0.8735 | 8.2 | 0.0206 | 6.4 | 0.2003 | 1.0 | 0.2126 | 0.7 | 3.4 |
| computation budget: 4 Models | | | | | | | | | | | | | |
| #958 | -22.05 | -21.27 | 0.5268 | -1.0 | 0.8744 | 8.4 | 0.0202 | 8.2 | 0.1887 | 6.8 | 0.2159 | -0.9 | **4.3** |
| #480 | -22.83 | -21.48 | 0.5168 | 0.9 | 0.8734 | 8.2 | 0.0210 | 4.5 | 0.2018 | 0.3 | 0.2146 | -0.3 | 2.7 |
| #360 | -22.05 | -21.27 | 0.5178 | 0.7 | 0.8735 | 8.2 | 0.0206 | 6.4 | 0.2003 | 1.0 | 0.2126 | 0.7 | 3.4 |
| #1037 | -41.93 | -41.27 | 0.5300 | -1.6 | 0.8725 | 8.1 | 0.0212 | 3.6 | 0.1888 | 6.7 | 0.2192 | -2.4 | 2.9 |
| #962 | -28.24 | -27.68 | 0.5124 | 1.8 | 0.8739 | 8.3 | 0.0204 | 7.3 | 0.1920 | 5.1 | 0.2184 | -2.1 | 4.1 |
| computation budget: 3 Models | | | | | | | | | | | | | |
| #1037 | -41.93 | -41.27 | 0.5300 | -1.6 | 0.8725 | 8.1 | 0.0212 | 3.6 | 0.1888 | 6.7 | 0.2192 | -2.4 | 2.9 |
| #1046 | -41.93 | -41.27 | 0.5368 | -2.9 | 0.8723 | 8.1 | 0.0201 | 8.6 | 0.1987 | 1.8 | 0.2118 | 1.0 | **3.3** |
| #943 | -40.13 | -40.01 | 0.5308 | -1.7 | 0.8746 | 8.4 | 0.0208 | 5.5 | 0.1998 | 1.3 | 0.2101 | 1.8 | 3.0 |
| #1063 | -48.11 | -47.68 | 0.5488 | -5.2 | 0.8730 | 8.2 | 0.0210 | 4.5 | 0.1897 | 6.3 | 0.2207 | -3.1 | 2.1 |
| #479 | -40.91 | -40.22 | 0.5407 | -3.6 | 0.8724 | 8.1 | 0.0198 | 10.0 | 0.2040 | -0.8 | 0.2132 | 0.4 | 2.8 |
| computation budget: 2 Models | | | | | | | | | | | | | |
| #817 | -60.00 | -60.00 | 0.5891 | -12.9 | 0.8725 | 8.1 | 0.0200 | 9.1 | 0.1915 | 5.4 | 0.2105 | 1.6 | **2.3** |
| #562 | -60.00 | -60.00 | 0.6216 | -19.1 | 0.8713 | 8.0 | 0.0202 | 8.2 | 0.1976 | 2.4 | 0.2011 | 6.0 | 1.1 |
| #4697 | -60.13 | -60.01 | 0.5985 | -14.7 | 0.8734 | 8.2 | 0.0202 | 8.2 | 0.1983 | 2.0 | 0.2085 | 2.6 | 1.3 |
| #6539 | -60.91 | -60.21 | 0.5911 | -13.3 | 0.8705 | 7.9 | 0.0217 | 1.4 | 0.1959 | 3.2 | 0.2062 | 3.6 | 0.6 |
| #1 | -60.00 | -60.00 | 0.6205 | -18.9 | 0.8692 | 7.7 | 0.0205 | 6.8 | 0.2018 | 0.3 | 0.2150 | -0.5 | -0.9 |
| computation budget: 1 Model | | | | | | | | | | | | | |
| #0 | -80.00 | -80.00 | 0.5994 | -14.9 | 0.8390 | 4.0 | 0.0265 | -20.5 | 0.1947 | 3.8 | 0.2072 | 3.2 | -4.9 |

## I  Recommended Tree-Structured Multi-Task Model Architectures

As introduced in Section 4.1, we conduct experiments on NYUv2 (3 tasks) and Taskonomy (5 tasks) with Deeplab-ResNet34 (5 branching points) and MobileNetV2 (5 branching points). Figure 5 and 6 show the specific structures of the multi-task models recommended by our framework. For simplicity, the model architectures are depicted by their equivalent layouts.

## J  More Comparisons with State-of-the-Art Methods

Table 9 ∼ 11 report more comparisons with state-of-the-art MTL methods. Compared with general MTL methods, our recommended model outperforms existing works in computation cost and the number of parameters with competitive task performance. Compared with branched MTL methods, What-to-Share, BMTAS, and Task-Grouping, in terms of the overall task performance in the $\Delta t$ columns, our top-1 multi-task architectures could achieve higher results (5.2% *vs* 4.8%/4.9%, 3.1%, *vs* -0.5%/-1.0%/0.9%), which indicates that our recommender has the ability to search out predominant architectures in the tree-structured multi-task model design space.

$$\begin{bmatrix} [(1,2,3)] \\ [(1,2,3)] \\ [(1,2,3)] \\ [(1,2,3)] \\ [(1,2,3)] \end{bmatrix} \quad \begin{bmatrix} [(1,2,3)] \\ [(1,2,3)] \\ [(1,2,3)] \\ [(1,2),(3)] \\ [(1,2),(3)] \end{bmatrix} \quad \begin{bmatrix} [(1,2,3)] \\ [(1,2,3)] \\ [(1,2,3)] \\ [(1,2,3)] \\ [(1,2),(3)] \end{bmatrix} \quad \begin{bmatrix} [(1,2,3)] \\ [(1,2,3)] \\ [(1,2),(3)] \\ [(1,2),(3)] \\ [(1,2),(3)] \end{bmatrix} \quad \begin{bmatrix} [(1,2,3)] \\ [(1,2,3)] \\ [(1,2,3)] \\ [(1,2,3)] \\ [(1,3),(2)] \end{bmatrix}$$
$$\quad\#0 \qquad\qquad \#45 \qquad\qquad \#50 \qquad\qquad \#37 \qquad\qquad \#49$$

(a) Corresponding to Table 1 on Deeplab-ResNet34.

$$\begin{bmatrix} [(1,2,3)] \\ [(1,2,3)] \\ [(1,2,3)] \\ [(1,2,3)] \\ [(1,2,3)] \end{bmatrix} \begin{bmatrix} [(1,3),(2)] \\ [(1,3),(2)] \\ [(1,3),(2)] \\ [(1,3),(2)] \\ [(1,3),(2)] \end{bmatrix} \begin{bmatrix} [(1,3),(2)] \\ [(1,3),(2)] \\ [(1,3),(2)] \\ [(1,3),(2)] \\ [(1),(3),(2)] \end{bmatrix} \begin{bmatrix} [(1,3),(2)] \\ [(1,3),(2)] \\ [(1,3),(2)] \\ [(1),(3),(2)] \\ [(1),(3),(2)] \end{bmatrix} \begin{bmatrix} [(1,3),(2)] \\ [(1,3),(2)] \\ [(1),(3),(2)] \\ [(1),(3),(2)] \\ [(1),(3),(2)] \end{bmatrix}$$
$$\#0 \qquad\qquad \#7 \qquad\qquad \#11 \qquad\qquad \#10 \qquad\qquad \#9$$

(b) Corresponding to Table 6 on MobileNetV2.

Figure 5: Recommended multi-task models on NYUv2.

$$\begin{bmatrix} [(1),(2,3,4,5)] \\ [(1),(2,3),(4,5)] \\ [(1),(2,3),(4),(5)] \\ [(1),(2,3),(4),(5)] \\ [(1),(2),(3),(4),(5)] \end{bmatrix} \begin{bmatrix} [(1,4),(2,3,5)] \\ [(1,4),(2,3),(5)] \\ [(1,4),(2,3),(5)] \\ [(1,4),(2),(3),(5)] \\ [(1,4),(2),(3),(5)] \end{bmatrix} \begin{bmatrix} [(1,4),(2,3,5)] \\ [(1,4),(2,3,5)] \\ [(1,4),(2,3,5)] \\ [(1,4),(2,5),(3)] \\ [(1,4),(2,5),(3)] \end{bmatrix} \begin{bmatrix} [(1,4),(2,3,5)] \\ [(1,4),(2,3,5)] \\ [(1,4),(2,3,5)] \\ [(1,4),(2,3,5)] \\ [(1,4),(2,3,5)] \end{bmatrix}$$
$$\#353 \qquad\qquad \#958 \qquad\qquad \#1046 \qquad\qquad \#817$$

(a) Corresponding to Table 2 on Deeplab-ResNet34.

$$\begin{bmatrix} [(3),(1,2,4,5)] \\ [(3),(1,2,4,5)] \\ [(3),(1,2,4,5)] \\ [(3),(1,4),(2,5)] \\ [(3),(1,4),(2,5)] \end{bmatrix} \begin{bmatrix} [(3),(1,2,4,5)] \\ [(3),(1,2,4,5)] \\ [(3),(1,2,4,5)] \\ [(3),(1,5),(2,4)] \\ [(3),(1),(2,4),(5)] \end{bmatrix} \begin{bmatrix} [(3),(1,2,4,5)] \\ [(3),(1,2,4,5)] \\ [(3),(1,2,4,5)] \\ [(3),(1,2,4,5)] \\ [(3),(1,2,4,5)] \end{bmatrix} \begin{bmatrix} [(3),(1,2,4,5)] \\ [(3),(1,2,4,5)] \\ [(3),(1,2,4,5)] \\ [(3),(1,2,4,5)] \\ [(3),(1,5),(2,4)] \end{bmatrix} \begin{bmatrix} [(3),(1,2,4,5)] \\ [(3),(1,2,4,5)] \\ [(3),(1,2,4,5)] \\ [(3),(1,2,4,5)] \\ [(3),(1,2,4),(5)] \end{bmatrix}$$
$$\#3221 \qquad\qquad \#3220 \qquad\qquad \#2947 \qquad\qquad \#3215 \qquad\qquad \#3261$$

(b) Corresponding to Table 7 on MobileNetV2.

Figure 6: Recommended multi-task models on Taskonomy.

Table 9: Comparison with state-of-the-art MTL methods for NYUv2 using Deeplab-ResNet34.

| Model | FLOPs (%) ↓ | #Params (%) ↓ | Semantic Seg. mIoU ↑ | Pixel Acc. ↑ | $\Delta t_1$ ↑ | Surface Normal Prediction Error Mean ↓ | Median ↓ | $\theta$, within 11.25° ↑ | 22.5° ↑ | 30° ↑ | $\Delta t_2$ ↑ | Depth Estimation Error Abs. ↓ | Rel. ↓ | $\delta$, within 1.25 ↑ | $1.25^2$ ↑ | $1.25^3$ ↑ | $\Delta t_3$ ↑ | $\Delta t$ ↑ |
|---|---|---|---|---|---|---|---|---|---|---|---|---|---|---|---|---|---|---|
| Ind. Models | - | - | 26.50 | **58.20** | - | 17.70 | 16.30 | 29.40 | 72.30 | 87.30 | - | 0.62 | 0.24 | 57.80 | 85.80 | 96.00 | - | - |
| What-to-Share | -1.54 | -0.38 | 25.66 | 57.64 | -2.1 | 17.75 | 16.38 | 29.15 | **73.02** | **87.44** | -0.1 | 0.60 | 0.22 | 60.49 | 87.45 | 96.55 | 3.7 | 0.5 |
| Cross-Stitch | 0.00 | 0.00 | 25.4 | 57.6 | -2.6 | 17.2 | 14.0 | 41.4 | 67.7 | 80.4 | 8.7 | 0.58 | 0.23 | 61.4 | 88.4 | 95.5 | 3.9 | 3.3 |
| Sluice | 0.00 | 0.00 | 23.8 | 56.9 | -6.2 | 17.2 | 14.4 | 38.9 | 69.0 | 81.4 | 7.1 | 0.58 | 0.24 | 61.9 | 88.1 | 96.3 | 3.3 | 1.4 |
| NDDR-CNN | 6.44 | 5.00 | 21.6 | 53.9 | -12.9 | **17.1** | 14.5 | 37.4 | 70.9 | 83.1 | 7.0 | 0.66 | 0.26 | 55.7 | 83.7 | 94.8 | -4.4 | -3.5 |
| MTAN | 22.11 | 3.70 | 26.0 | 57.2 | -1.8 | 17.2 | 13.9 | **43.7** | 70.5 | 81.9 | **11.5** | 0.57 | 0.25 | 62.7 | 87.7 | 95.9 | 2.9 | 4.2 |
| DEN | 10.81 | -62.70 | 23.9 | 54.9 | -7.7 | **17.1** | 14.8 | 36.0 | 70.6 | 83.4 | 5.6 | 0.97 | 0.31 | 22.8 | 62.4 | 88.2 | -36.3 | -12.8 |
| AdaShare | -6.24 | **-66.67** | 24.4 | 57.8 | -4.3 | 17.7 | **13.8** | 42.3 | 68.9 | 80.5 | 9.3 | 0.59 | **0.20** | 61.3 | 88.5 | 96.5 | 6.2 | 3.8 |
| AutoMTL | -0.45 | -45.10 | **26.6** | 58.2 | **0.2** | 17.3 | 14.4 | 39.1 | 70.7 | 83.1 | 8.0 | **0.54** | 0.22 | **65.1** | **89.2** | **96.9** | 7.8 | **5.3** |
| Top-1 | -66.67 | -66.67 | 25.23 | 57.69 | -2.8 | 17.14 | 15.15 | 35.85 | 72.20 | 85.54 | 6.0 | 0.55 | 0.23 | 63.85 | 89.38 | 97.03 | 5.8 | 3.0 |

Table 10: Comparison with Branched MTL methods for NYUv2 using MobileNetV2.

| Model | FLOPs (%)↓ | #Params (%)↓ | mIoU↑ | Pixel Acc.↑ | Δt₁↑ | Mean | Median | 11.25° | 22.5° | 30° | Δt₂↑ | Abs. | Rel. | 1.25 | 1.25² | 1.25³ | Δt₃↑ | Δt↑ |
|---|---|---|---|---|---|---|---|---|---|---|---|---|---|---|---|---|---|---|
| | | | Semantic Seg. | | | Surface Normal Prediction | | | | | | Depth Estimation | | | | | | |
| | | | | | | Error↓ | | θ, within↑ | | | | Error↓ | | δ, within↑ | | | | |
| Ind. Models | - | - | 20.36 | **49.44** | - | 18.17 | 16.62 | 28.37 | 70.20 | 85.58 | - | 0.77 | 0.28 | 47.92 | 78.46 | 92.81 | - | - |
| What-to-Share | -8.41 | -0.30 | **21.10** | 49.03 | 1.4 | 17.82 | **15.71** | 30.61 | 72.60 | 86.08 | 3.9 | 0.67 | 0.25 | 54.71 | 83.25 | 94.97 | 9.3 | 4.8 |
| BMTAS | -64.46 | -33.41 | 18.98 | 48.40 | -4.4 | **17.71** | 16.09 | 29.74 | **72.70** | **86.90** | 3.1 | **0.60** | 0.24 | **60.73** | **87.25** | 96.33 | **15.9** | 4.9 |
| **Top-1** | **-66.67** | **-66.67** | 19.36 | 48.97 | -2.9 | 17.99 | 16.02 | **31.43** | 70.41 | 84.65 | 2.9 | 0.61 | **0.23** | 60.02 | 86.89 | **96.34** | 15.7 | **5.2** |

Table 11: Comparison with Branched MTL methods for Taskonomy using MobileNetV2.

| Models | FLOPs (%)↓ | #Params (%)↓ | Semantic Seg. Abs.↓ | Δt₁↑ | Normal Pred. Abs.↑ | Δt₂↑ | Depth Est. Abs.↓ | Δt₃↑ | Keypoint Det. Abs.↓ | Δt₄↑ | Edge Det. Abs.↓ | Δt₅↑ | Δt↑ |
|---|---|---|---|---|---|---|---|---|---|---|---|---|---|
| Ind. Models | - | - | 0.5217 | - | 0.807 | - | 0.022 | - | 0.2024 | - | 0.214 | - | - |
| What-to-Share | -5.02 | -0.18 | 1.0283 | -1.9 | 0.7656 | -0.1 | 0.0275 | **0.7** | 0.2417 | -0.9 | 0.2688 | -0.3 | -0.5 |
| BMTAS | **-78.47** | **-76.32** | 1.0239 | -1.4 | 0.7511 | -2.0 | 0.0322 | -16.2 | 0.2202 | **8.1** | 0.2508 | **6.5** | -1.0 |
| Task-Grouping | -25.11 | -5.21 | 0.9965 | 1.3 | 0.7678 | **0.2** | 0.0287 | -3.6 | 0.2323 | 3.0 | 0.2591 | 3.4 | 0.9 |
| **Top-1** | -53.99 | -44.86 | 0.977 | **3.2** | 0.7625 | -0.5 | 0.0277 | 0.0 | 0.2232 | 6.8 | 0.2519 | 6.0 | **3.1** |

(b) Did you describe the limitations of your work? [Yes]

(c) Did you discuss any potential negative societal impacts of your work? [Yes] See Section 5.

(d) Have you read the ethics review guidelines and ensured that your paper conforms to them? [Yes]

2. If you are including theoretical results…

    (a) Did you state the full set of assumptions of all theoretical results? [Yes] See Section 3.1 and Appendix Section B and C.

    (b) Did you include complete proofs of all theoretical results? [Yes] See Appendix Section E.

3. If you ran experiments…

    (a) Did you include the code, data, and instructions needed to reproduce the main experimental results, including all requirements (e.g., `requirements.txt` with explicit version), an instructive README with installation, and execution commands (either in the supplemental material or as a URL)? [Yes] We describe data, models, and experiments in detail. [No] We have not shared code when this answer is written, but plan to open-source the code to assist future research.

    (b) Did you include the raw results of running the given instructions on the given code and data? [N/A] We will include them in the future public source.

    (c) Did you include scripts and commands that can be used to generate the figures and tables in your paper based on the raw results of the code, data, and instructions given? [N/A] We will include them in the future public source.

    (d) Did you ensure sufficient code quality such that your code can be safely executed and the code is properly documented? [N/A] We will ensure it in the future public source.

    (e) Did you specify all the training details (e.g., data splits, pre-processing, search spaces, fixed hyperparameter settings, and how they were chosen)? [Yes] See Section 4.1.

(f) Did you ensure that you compared different methods (including your own) exactly on the same benchmarks, including the same datasets, search space, code for training and hyperparameters for that code? [Yes] See Section 4.1 and Appendix Section G.

(g) Did you run ablation studies to assess the impact of different components of your approach? [N/A] Our approach consists of three components and cannot work without any one of them. In the paper, we verified the effectiveness of each component through experiments or theoretical proofs.

(h) Did you use the same evaluation protocol for the methods being compared? [Yes] See Section 4.1.

(i) Did you compare performance over time? [No]

(j) Did you perform multiple runs of your experiments and report random seeds? [Yes] See Section 4.3.

(k) Did you report error bars (e.g., with respect to the random seed after running experiments multiple times)? [No] Instead we conduct correlation experiments to show the reliability of our recommender for each dataset with different backbone models and random seeds.

(l) Did you use tabular or surrogate benchmarks for in-depth evaluations? [No]

(m) Did you include the total amount of compute and the type of resources used (e.g., type of GPUs, internal cluster, or cloud provider)? [No] Our algorithms can be executed on any computing resources. In this paper, we train the associated 2-task models on NVIDIA 1080ti and the recommended multi-task models on NVIDIA m40 in parallel.

(n) Did you report how you tuned hyperparameters, and what time and resources this required (if they were not automatically tuned by your AutoML method, e.g. in a NAS approach; and also hyperparameters of your own method)? [N/A] We conduct experiments with fixed hyperparameters following prior works.

4. If you are using existing assets (e.g., code, data, models) or curating/releasing new assets…

(a) If your work uses existing assets, did you cite the creators? [Yes] See Section 4.1.

(b) Did you mention the license of the assets? [N/A] Our experiments were conducted on publicly available datasets.

(c) Did you include any new assets either in the supplemental material or as a URL? [No] We did not introduce new datasets.

(d) Did you discuss whether and how consent was obtained from people whose data you're using/curating? [No] Our experiments were conducted on publicly available datasets.

(e) Did you discuss whether the data you are using/curating contains personally identifiable information or offensive content? [No] We are not aware of relevant issues in the data we use.

5. If you used crowdsourcing or conducted research with human subjects…

(a) Did you include the full text of instructions given to participants and screenshots, if applicable? [N/A] We didn't use crowdsourcing.

(b) Did you describe any potential participant risks, with links to Institutional Review Board (IRB) approvals, if applicable? [N/A] We didn't use crowdsourcing.

(c) Did you include the estimated hourly wage paid to participants and the total amount spent on participant compensation? [N/A] We didn't use crowdsourcing.