# OpenReview forum: "A Tree-Structured Multi-Task Model Recommender"
_automl.cc/AutoML/2022/Track/Main — AutoML-Conf 2022 (Main Track)_

### Official Review · Reviewer_Utff · 2022-03-31

**Potential Impact On The Field Of Automl:** N/A for reproducibility reviewers
**Potential Impact On The Field Of Automl Rating:** 3
**Technical Quality And Correctness:** N/A for reproducibility reviewers
**Technical Quality And Correctness Rating:** 3
**Clarity:** N/A for reproducibility reviewers
**Clarity Rating:** 3

**Summary Of Contributions:**

N/A for reproducibility reviewers

**Overall Review:**

N/A for reproducibility reviewers

**Reproducibility:**

The authors have not shared code but plan to open-source the code to assist future research.
Although the data&models&experiments are described in detail, it cannot be reproduced due to the lack of Code Supplement.

**Review Confidence:**

4: You are confident in your assessment, but not absolutely certain. It is unlikely, but not impossible, that you did not understand some parts of the submission or that you are unfamiliar with some pieces of related work.

**Review Rating:**

4: Marginally above the acceptance threshold (use sparsely)

**Review Summary:**

N/A for reproducibility reviewers

---

### Official Review · Reviewer_UkSU · 2022-04-03

**Potential Impact On The Field Of Automl Rating:** 3
**Technical Quality And Correctness Rating:** 4
**Clarity Rating:** 4

**Summary Of Contributions:**

The authors work with Multi-task learning, which aims to solve multiple tasks simultaneously. The authors propose an approach that takes as inputs a backbone model and a set of tasks in interest and then predicts the top-k tree-structured multi-task architectures that achieve high task accuracy while meeting a user-specified computation budget. The authors propose to achieve it using three major components: a task accuracy estimator, design space enumerator, and branching point predictor.

Unlike previous approaches, the authors claim that their approach does branching automatically, without the need for domain expertise, and efficient computation, and can work under a given constraint.
The authors furthermore maintain their assertions from the experiments on various datasets.

**Clarity:**

The paper is clearly written and is easy to follow. Readers can easily capture the main ideas (as we mentioned in item 1).



**Overall Review:**

## Strengths

* The paper proposes a novel approach for MTL using a tree-structured architecture candidate generation.
* The authors show the robustness of this approach across multiple datasets and settings.

## Weakness

* The authors propose a two-task accuracy estimator. However, it would be nice to share how the task interactions and interference would be if we did not consider this scenario. Why not more than 2? Is it for complexity reasons? Or does this setting achieve best performance.
* The models have only been run on one seed, and it would be very useful to see it run on multiple seeds, reporting both mean and var of performance, to show that their approach is not an artifact of luck, and is robust.

**Potential Impact On The Field Of Automl:**

The work performed by the authors does have a significant impact and is of interest to the AutoML community in general. I have come to this conclusion for three reasons:
* The approach is end-to-end and can be deployed on any arbitrary pre-trained model.
* The approach fares well against current SOTA, and does not require any re-training.
* It gives the top-k candidate set of architectures that perform well for the given set of tasks and aid better analysis studies in the future.


**Reproducibility:**

The authors provide sufficient information to reproduce the experiments mentioned in the paper.



**Review Confidence:**

5: You are absolutely certain about your assessment. You are very familiar with the related work and checked all the details carefully.

**Review Rating:**

5: Accept, good paper

**Review Summary:**

Overall, I find the paper exciting and fun to read. The authors introduce concepts that help in the field of MTL and are very useful to the research community in general. Their end-to-end approach is efficient and does not require additional training, unlike many similar approaches. Furthermore, it returns a candidate set of architectures and makes the approach very useful for analyzing the neural network better.

**Technical Quality And Correctness:**

The paper is technically sound across most sections, but I do have one question I hope the authors could answer for me:

* **Section 3.1**: The authors propose the task accuracy estimator by averaging over all associated two-task architectures. I would suggest running with multiple seeds and reporting the variance as well, to confirm that the model consistently gives outputs that have a high correlation with the oracle ranking, and that the results are not an artifact of the seed.
* Why do you consider only two task architectures and not a subset of all task structures? Do we not need to worry about the task interactions and interventions here?

---

### Official Review · Reviewer_k5xU · 2022-04-05

**Potential Impact On The Field Of Automl Rating:** 3
**Technical Quality And Correctness Rating:** 3
**Clarity Rating:** 3

**Summary Of Contributions:**

1. This paper proposes a tree-structured multi-task model recommender. It takes as input an arbitrary CNN model and some predefined tasks, and outputs the top-k tree-structured multi-task architectures that achieve good performance within some computation budget limitation.

2. The whole process is in a white-box manner.

3. Extensive experiments validate the effectiveness of the proposed approach

**Clarity:**

The paper is well written and very easy to follow. Though there are some reference citing error, such as,
Line 145, Line 270, etc: Appendix Section ??


**Overall Review:**

Please see the above section: **Technical Quality And Correctness**

**Potential Impact On The Field Of Automl:**

Personally speaking, during the process, how to estimate the task accuracy without actual training is pretty interesting and might be useful for autoML community.



**Reproducibility:**

They describe data, models, and experiments in detail, though they have not shared code when this answer is written, but plan to open-source the code to assist future research.

I think it is difficult to reproduce the result without the authors' code considering the complexity of multi task learning in the paper.

**Review Confidence:**

3: You are fairly confident in your assessment. It is possible that you did not understand some parts of the submission or that you are unfamiliar with some pieces of related work.

**Review Rating:**

5: Accept, good paper

**Review Summary:**

This paper proposes a tree-structured multi-task model recommender. It takes as input an arbitrary CNN model and some predefined tasks, and outputs the top-k tree-structured multi-task architectures that achieve good performance within some computation budget limitation in a white-box manner.

The proposed approach consists of three key components:
- Branching point detector: automatically detects branching points (computation blocks, such as Residual Block in ResNet50)
- Design space enumerator: lists all the multitask architecture
- Task accuracy estimator: predicts task performance of architecture w/o performing actual training.

Please check the above section **Technical Quality And Correctness** for negative points.

**Technical Quality And Correctness:**

+The proposed approach, theory, and experiments are sound.

-Among the three key components: Branching point detector, Design space enumerator, and Task accuracy estimator, it is necessary to explain how task accuracy estimator (predicts task performance of architecture w/o performing actual training) works. For example:

- what is the motivation for SVDE instead of the regular entropy?
- why could the associated two-task models be used to evaluate the performance the actual multitask structure?
- how is the computation budget involved? This part is not clear. In Table 2, the column "Com.Budget" indicates the number of Models? Different models might have varying different number of parameters and the number of models could be used to represent budget?

---

### Meta-Review · Area_Chair_hsCC · 2022-05-09

**Recommendation:** Accept
**Confidence:** 4

**Metareview:**

This paper proposes a recommender that, given a convolutional model and a set of tasks, proposes (top-k) tree-structured multi-task architectures that can achieve high accuracy (by figuring out where to branch so that the appropriate parts of the network are shared across tasks) while adhering to a computational budget. The reviewers found the proposed approach, experiments and theory sound and novel, the presentation clear, the empirical performance strong, and the contribution overall important for the research community. This is why I recommend acceptance.

The reviewers though do point out some flaws that should be addressed as well as possible in the revised version. These relate to motivating components of the approach, clarity with regards to measuring computational budget, and the need to run experiments with additional random seeds.

An important thing to flag is that while the authors claim they will publish their code on Github as soon as possible, as it stands currently their code is not available.

---

### Decision · Program_Chairs · 2022-05-13

Accept